# Maternal Iodine Status and Birth Outcomes: A Systematic Literature Review and Meta-Analysis

**DOI:** 10.3390/nu15020387

**Published:** 2023-01-12

**Authors:** Darren C. Greenwood, James Webster, Claire Keeble, Elizabeth Taylor, Laura J. Hardie

**Affiliations:** 1Leeds Institute of Cardiovascular and Metabolic Medicine, School of Medicine, University of Leeds, Leeds LS2 9JT, UK; 2Leeds Institute for Data Analytics, University of Leeds, Leeds LS2 9JT, UK; 3School of Food Science and Nutrition, University of Leeds, Leeds LS2 9JT, UK

**Keywords:** Iodine, birth weight, infant, small for gestational age, meta-analysis

## Abstract

Background & aims: Iodine is important for thyroid function during pregnancy to support fetal growth, but studies of maternal iodine status and birth outcomes are conflicting. We aimed to quantify the association between iodine status and birth outcomes, including potential threshold effects using nonlinear dose–response curves. Methods: We systematically searched Medline and Embase to 10 October 2022 for relevant cohort studies. We conducted random-effects meta-analyses of urinary iodine concentration (UIC), iodine:creatinine ratio (I:Cr), and iodide intake for associations with birth weight, birth weight centile, small for gestational age (SGA), preterm delivery, and other birth outcomes. Study quality was assessed using the Newcastle-Ottawa scale. Results: Meta-analyses were conducted on 23 cohorts with 42269 participants. Birth weight was similar between UIC ≥ 150 μg/L and <150 μg/L (difference = 30 g, 95% CI −22 to 83, *p* = 0.3, *n* = 13, I^2^ = 89%) with no evidence of linear trend (4 g per 50 μg/L, −3 to 10, *p* = 0.2, *n* = 12, I^2^ = 80%). I:Cr was similar, but with nonlinear trend suggesting I:Cr up to 200 μg/g associated with increasing birthweight (*p* = 0.02, *n* = 5). Birthweight was 2.0 centiles (0.3 to 3.7, *p* = 0.02, *n* = 4, I^2^ = 0%) higher with UIC ≥ 150 μg/g, but not for I:Cr. UIC ≥ 150 μg/L was associated with lower risk of SGA (RR = 0.85, 0.75 to 0.96, *p* = 0.01, *n* = 13, I^2^ = 0%), but not with I:Cr. Conclusions: The main risk of bias was adjustment for confounding, with variation in urine sample collection and exposure definition. There were modest-sized associations between some measures of iodine status, birth weight, birth weight centile, and SGA. In pregnancy, we recommend that future studies report standardised measures of birth weight that take account of gestational age, such as birth weight centile and SGA. Whilst associations were modest-sized, we recommend maintaining iodine sufficiency in the population, especially for women of childbearing age on restricted diets low in iodide.

## 1. Introduction

Iodine has an important role in normal thyroid function, with demands for iodine increasing during pregnancy to support fetal growth as well as compensating for increased renal clearance [1,2]. Severe maternal iodine deficiency is considered an established risk factor for maternal goitre and neurological impairment in the neonate [1,3], with possible associations with subsequently less developed motor skills and intellectual capacity [4]. To avoid these outcomes, pregnant populations are defined by the World Health Organization (WHO) as having insufficient iodine where the median urinary iodine concentration (UIC) is less than 150 μg/L [1,3].

Despite salt iodisation programmes available across many regions, the populations of over 50 countries are still considered to be affected by iodine deficiency [5]. To avoid iodine deficiency, WHO recommend an iodide intake of 150 μg/day in adults, and 250 μg/day during pregnancy [3]. However, two thirds of European countries that monitor iodine in pregnancy have reported inadequate iodine intake [6].

Dairy and seafood are the main dietary sources of dietary iodine, excluding supplements and fortified foods such as iodised salt. With recent increases in diets that restrict intake of these foods, particularly in women of childbearing age, the iodine status of populations previously considered sufficient are moving more towards mild deficiency [7,8]. Whilst the effects of severe deficiency are known, the extent to which adverse outcomes are associated with milder levels of deficiency is less clear.

Two recent systematic reviews of observational studies of iodine and birth outcomes drew differing conclusions [9,10]. Since these reviews were published, results on iodine and birth outcomes from a number of large birth cohorts have become available, including the Norwegian Mother and Child Cohort Study (MoBa) [11], the Screening for Pregnancy Endpoints (SCOPE) cohort [12], the Born in Bradford (BiB) cohort [13], and results from the Shanghai birth registry [14]. There is potential for studies in deficient populations to draw substantially different conclusions from those in less-deficient populations, and for effects to only be seen in individuals with particularly poor intakes. We therefore aimed to systematically review the new and existing literature, additionally investigating potential threshold effects.

## 2. Material and Methods

### 2.1. Search Strategy

We conducted a systematic literature review of cohort studies that included relevant information on both maternal iodine status and birth outcomes. We searched MEDLINE and EMBASE databases through OVID, up to 10 October 2022 using a PICO structure, with a cohort study search filter, synonyms relating to pregnancy, iodine status, and specific birth outcomes, with adjacency terms and allowing for alternative spellings where appropriate. The key words included synonyms relating to pregnancy, maternity, and birth (study population); iodide, iodine, and urinary biomarkers of iodine (exposure); and birthweight, fetal growth, head circumference, preterm, and spontaneous abortion (outcomes). The complete search strategies are detailed in Appendix A).

We included preferred reporting items for systematic reviews and meta-analyses (PRISMA) and followed guidelines for conducting meta-analysis of observational studies in epidemiology (MOOSE) throughout the review. The review protocol was published on PROSPERO (registration number CRD42016043748) prior to starting the study.

### 2.2. Study Selection

Eligible studies were cohorts, case-control studies nested within cohorts, and case-cohort studies, with exposure measured during pregnancy. Relevant exposures were urinary iodine concentration (UIC), urinary iodine excretion or iodine to creatinine ratio (I:Cr), and iodide intake. Relevant outcomes were pre-specified as those related to fetal loss, preterm birth, and birth size. Low birth weight and macrosomia were taken as birth weight <2.5 kg and >4 kg, respectively.

Birth weight centile was based on standardized birth weight adjusting for at least gestational age, using the study-specific definitions, and transforming z-scores to centiles where necessary. Small for gestational age (SGA) was defined as birth weight <10th centile and large for gestational age (LGA) >90th centile. Spontaneous abortion was defined as fetal loss ≤24 weeks gestation and stillbirth as fetal loss >24 weeks gestation. Preterm delivery was taken as spontaneous preterm birth <37 weeks, or any preterm delivery <37 weeks if this information was not available.

### 2.3. Data Extraction and Quality Assessment

Titles and abstracts were screened by two independent reviewers (from CK, ET, JW, LJH) using Screenatron and Disputatron [15], with disagreements resolved by a third reviewer (DCG). Full text screening was conducted by two independent reviewers (DCG, LJH) with disagreements resolved by consensus. Data were extracted by DCG and numeric data independently checked by JW. Where two publications reported results from the same cohorts, data were extracted from the most complete report. Information was also extracted from published study protocols or cohort profiles where necessary. For three studies, investigators were contacted for additional data [12,13,16]. Non-English language and unpublished articles and abstracts were excluded.

We assessed methodological quality of studies using the Newcastle–Ottawa scale for cohort studies [17]. Studies were rated for representativeness of the exposed and non-exposed cohorts, ascertainment of exposure, and the outcome not being present before the study. Levels of exposure were considered comparable if length of gestation was controlled for (apart from for preterm deliveries), either through adjustment or by outcome definition, and adjustment for any other relevant confounder. Outcomes were rated on being objective, followed until the end of pregnancy, with >90% of potential participants following up. Quality was assessed independently by two reviewers (CK, ET) with disagreements resolved by a third (DCG) for the first six consecutive studies retrieved, after which all studies were assessed by DCG. All studies were included regardless of perceived quality, but risk of bias was taken into account in the interpretation.

### 2.4. Data Synthesis and Analysis

Data from each study were synthesised by meta-analysis, comparing dichotomous exposures of UIC < 150 μg/L with ≥150 μg/L based on the WHO criteria [3] and I:Cr < 150 μg/g with ≥150 μg/g, alongside linear and nonlinear dose-response trends.

Linear and nonlinear dose-response trends in relative risks for binary outcomes (low birth weight, macrosomia, SGA, LGA, spontaneous abortion, still birth, and preterm delivery) were derived using Greenland and Longnecker’s method [18,19]. The estimated mean iodine status for each category of exposure was extracted, or the midpoint of each category derived where the mean or median were not provided. For unbounded upper limits, we assumed the category width was 1.5 times the adjacent one. If the reference category was not the lowest, we used the Greenland and Longnecker fitted counts to express adjusted risks relative to the lowest category [18,20]. This method also allowed us to combine categories to estimate relative risks for dichotomised iodine status.

To compare dichotomised exposures, adjusted means were first combined using common (fixed) effect meta-analysis where necessary, then differences pooled using random-effects meta-analysis. Linear trends in continuous outcomes (birth weight, birth weight centile, birth length, and head circumference) were estimated using multivariate random-effects meta-analysis [21,22] and presented per 50 μg/L higher UIC or 50 μg/g higher I:Cr. Nonlinear trends for each study were estimated using restricted cubic splines fitted to each study using knots at the 10th, 50th, and 90th percentiles [23], and then pooled using multivariable random-effects meta-analysis, compared to a reference of 150 μg/L for UIC, 150 μg/g for I:Cr, and 150 μg/d for iodide intake [20]. Meta-analyses were only conducted where more than two studies reported on the same outcome.

Between-study heterogeneity was expressed as the range of study estimates, and as a percentage of total variation (I^2^) [24,25]. Potential heterogeneity was explored through a limited number of pre-defined subgroup analyses where sufficient data were available as follows: by mean week of gestation when the urine samples were provided, median UIC of study population, high income vs. low-or-middle income country, by any adjustment for potential confounding such as gestational age (other than for preterm delivery), and by Newcastle-Ottawa domain score. Potential small-study effects such as publication bias were explored through contour-enhanced funnel plots and Egger’s test, where more than 10 studies reported on the same outcome. All analyses were conducted in Stata version 17 [26].

## 3. Results

### 3.1. Literature Search

Two hundred and forty-seven unique references were identified by the literature search. Of these, 53 were identified as potentially relevant after screening of titles and abstracts (with 89% agreement between reviewers) and 26 were identified as relevant following reading of the full texts (with 80% agreement between reviewers). These publications reported on 24 cohorts containing 42,503 participants (Figure 1, Appendix A) [11,12,13,14,16,27,28,29,30,31,32,33,34,35,36,37,38,39,40,41,42,43,44,45].

Eighteen cohorts reported UIC exposure, seven used I:Cr, three dietary intake and one total intake including dietary and supplemental sources. Eleven (46%) studies were from higher income countries, 14 (58%) were from iodine-deficient populations with median UIC < 150 μg/L, including six (25%) with median UIC < 100 μg/L. Study characteristics are presented in Table 1.

Study quality was assessed in duplicate for six papers (25%) with 89% agreement between reviewers across all items, and the remainder extracted by one reviewer. Overall study quality was generally good, with results based on established birth cohorts using objective measures of exposures and outcomes. The main differences in risk of potential bias derived from adjustment for confounding (Appendix A).

### 3.2. Birth Weight Outcomes

Fifteen studies reported birth weight in relation to UIC. Comparisons of UIC < 150 μg/L with ≥150 μg/L were derived for 13 studies, linear trends for 12 studies, and nonlinear trends for 11 studies. One study did not quantify the UIC exposure so could not be pooled [45]. There was no evidence that UIC ≥ 150 μg/L was associated with greater birth weight than <150 μg/L (difference = 30 g, 95% CI −22 to 83, *p* = 0.3, *n* = 13, I^2^ = 89%), nor of a linear trend (4 g per 50 μg/L, −3 to 10, *p* = 0.2, *n* = 12, I^2^ = 80%). Between-study heterogeneity was high, but there was no evidence of nonlinearity (*p* = 0.2, *n* = 11) (Figure 2a–c).

Six studies reported birth weight in relation to I:Cr. There was no evidence that I:Cr ≥ 150 μg/g was associated with greater birth weight than <150 μg/g (difference = 22 g, −12 to 56, *p* = 0.2, *n* = 5, I^2^ = 0%), nor linear trend (3 g per 50 μg/g, −7 to 14, *p* = 0.5, *n* = 5, I^2^ = 88%). However, there was evidence of nonlinearity (*p* = 0.02, *n* = 5), with an association between I:Cr and increasing birth weight evident up to around 200 μg/g (Figure 2d–f).

There were too few studies reporting iodide intake with the same category boundaries to conduct meta-analysis on the dichotomous exposure. There was no evidence of a linear association between intake and birth weight (9 g per 50 μg/day, −8 to 26, *p* = 0.3, *n* = 3, I^2^ = 85%), but some evidence of nonlinearity (*p* < 0.001, *n* = 3) (Appendix A).

For the binary outcomes of low birth weight (<2.5 kg) and macrosomia (>4 kg) there was no evidence of any association with UIC (Appendix A) and too few studies reported this outcome to conduct meta-analysis for I:Cr or iodide intake.

### 3.3. Standardised Birth Weight Outcomes

For birth weight centile, UIC ≥150 μg/L was associated with 2.0 centiles higher birth weight than UIC < 150 μg/L (difference = 2.0 centiles, 0.3 to 3.7, *p* = 0.02, *n* = 4, I^2^ = 0%), with no evidence of a linear dose-response trend (0.5 centiles per 50 μg/L, −0.5 to 1.4, *p* = 0.3, *n* = 4, I^2^ = 86%), and no evidence of nonlinearity (*p* = 0.4, *n* = 4) (Figure 3a–c). There was no evidence that I:Cr ≥ 150 μg/g was associated with higher birth weight centiles than I:Cr < 150 μg/g (difference =0.8, −4.8, 6.5, *p* = 0.8, *n* = 3, I^2^ = 35%), nor evidence of a linear trend (0.5 centiles per 50 μg/g, −0.3 to 1.3, *p* = 0.2, *n* = 5, I^2^ = 70%). There was also no evidence of nonlinearity (*p* = 0.1, *n* = 4) (Figure 3e–f).

UIC was associated with lower risk of SGA, in terms of ≥150 μg/L vs. UIC < 150 μg/L (RR = 0.85, 0.75 to 0.96, *p* = 0.01, *n* = 13, I^2^ = 0%), but with no evidence of a linear trend (RR = 0.96 per 50 μg/L, 0.92 to 1.01, *p* = 0.1, *n* = 8, I^2^ = 52%). There was no evidence of nonlinearity (*p* = 0.6, *n* = 7) (Figure 4a–c). There was no evidence of the same association with I:Cr, either for dichotomised exposure (RR = 0.95, 0.70 to 1.28, *p* = 0.7, *n* = 5, I^2^ = 66%) or as a linear trend (RR = 0.98 per 50 μg/g, 0.92 to 1.05, *p* = 0.6, *n* = 5, I^2^ = 70%). However, there was evidence of nonlinearity, with the lowest risk around 150 μg/g and higher risks associated with both lower and higher I:Cr (*p* = 0.003, *n* = 4). There were sufficient studies of iodide intake and SGA only to investigate linear trend, where there was no evidence of an association (RR = 0.95 per 50 μg/d, 0.87 to 1.05, *p* = 0.3, I^2^ = 35%) (Appendix A).

There were too few studies reporting LGA to conduct any meta-analyses for this outcome.

### 3.4. Birth Length and Head Circumference

There was no evidence of an association between birth length and UIC as a dichotomous exposure (difference = 0.0 cm, −0.1 to 0.2, *p* = 0.6, *n* = 5, I^2^ = 0%), as a linear trend (0.01 cm per 50 μg/L, 0.00 to 0.02, *p* = 0.1, *n* = 4, I^2^ = 0%), or any evidence of nonlinearity (*p* = 0.5, *n* = 5) (Appendix A). There were not enough data to conduct meta-analysis for I:Cr or iodide intake.

There was no evidence of an association between head circumference and UIC as a dichotomous exposure (difference = 0.0 cm, −0.2 to 0.2, *p* = 0.9, *n* = 5, I^2^ = 78%), as a linear trend (0.00 cm per 50 μg/L, −0.03 to 0.03, *p* = 0.8, *n* = 6, I^2^ = 16%), or any evidence of nonlinearity (*p* = 0.4, *n* = 6) (Appendix A). Neither was there evidence of an association between head circumference and I:Cr as a dichotomous exposure (difference = 0.1 cm, −0.1 to 0.3, *p* = 0.4, *n* = 3, I^2^ = 0%), as a linear trend (0.01 cm per 50 μg/g, −0.03 to 0.05, *p* = 0.6, *n* = 3, I^2^ = 49%), or any evidence of nonlinearity (*p* = 0.6, *n* = 3) (Appendix A). There were insufficient data to conduct meta-analyses for iodide intake.

### 3.5. Pregnancy Outcomes

Preterm delivery was defined as spontaneous preterm birth in only two studies [11,12], with the remainder using the broader definition of any delivery <37 weeks gestation. There was no evidence of an association between preterm delivery and UIC dichotomised into ≥150 μg/L vs. UIC < 150 μg/L (RR = 0.88, 0.72 to 1.08, *p* = 0.2, *n* = 12, I^2^ = 47%), or as a linear trend (RR = 0.97 per 50 μg/L, 0.92 to 1.02, *p* = 0.2, *n* = 12, I^2^ = 56%). There was also no evidence of nonlinearity in the trend (*p* = 0.1, *n* = 10) (Figure 5a–c). Similarly for preterm delivery and I:Cr, there was no evidence of an association with dichotomised I:Cr (RR = 0.97, 0.75 to 1.25, *p* = 0.8, *n* = 4, I^2^ = 14%) or as a linear trend (RR = 1.01 per 50 μg/g, 0.96 to 1.07, *p* = 0.7, *n* = 4, I^2^ = 0%). There was also no evidence of nonlinearity (*p* = 0.1, *n* = 4) (Figure 5d–f). There were sufficient studies of iodide intake and preterm delivery only to investigate linear trend, where there was no evidence of an association (RR = 0.99 per 50 μg/d, 0.95 to 1.04, *p* = 0.8, I2 = 0%) (Appendix A).

There were insufficient studies categorising UIC consistently to dichotomise into ≥150 μg/L and UIC < 150 μg/L for meta-analysis of spontaneous abortion ≤24 weeks gestation, however there was no evidence of a linear trend (RR = 0.88, 0.71 to 1.10, *p* = 0.3, *n* = 3, I^2^ = 94%), albeit with large between-study heterogeneity (Appendix A). There were insufficient studies to explore potential nonlinearity. There were not enough studies to conduct meta-analysis for I:Cr or iodide intake of this outcome.

There were not enough data to conduct meta-analyses for stillbirth as fetal loss >24 weeks gestation in relation to any outcome.

### 3.6. Subgroup Analyses and Small-Study Effects

There was no evidence that associations between birth weight and UIC differed by timing of urine collection, median population UIC, income status of country, or any adjustment for potential confounding (Appendix A, Appendix A). Studies with lower scores for selection had higher estimates comparing UIC ≥ 150 μg/L and <150 μg/L, but not with any other domains (Appendix A). There was no evidence of associations between SGA and UIC differing by subgroup, other than stronger linear trends seen in studies with lower Newcastle-Ottawa selection scores (Appendix A, Appendix A). For preterm delivery associations between higher UIC and lower risk of preterm delivery were stronger in low and middle-income countries in analyses of dichotomous exposures (*p* = 0.01) and linear trends (*p* = 0.02), and some differences between Newcastle-Ottawa comparability scores (Appendix A, Appendix A). There were too few studies reporting I:Cr or iodide intake to conduct subgroup analyses for these outcomes.

There was no evidence of any small-study effects such as publication bias for UIC and birth weight or preterm delivery, where sufficient studies existed to investigate, with no evidence of funnel plot asymmetry (Egger’s tests *p* = 0.4 and *p* = 0.9) (Appendix A).

## 4. Discussion

Our systematic review and series of meta-analyses have included nearly four times as many studies as the most recent reviews, involving analysis of iodine and birth outcomes for more than five times as many participants [9,10]. In addition to UIC, we have also included alternative measures of exposure, such as iodine to creatinine ratio and iodide intake. Moreover, we have investigated continuous dose-response trends over the full range of intakes, including both linear and potential nonlinear trends, to identify potential threshold or plateau effects.

There was evidence of children born to mothers with UIC > 150 μg/L having higher birth weight centile, and lower risk of SGA, but the association between these outcomes and I:Cr was less clear. We also found evidence of an association between higher I:Cr and higher birth weight, up to a threshold of around 200 μg/g, but not for UIC. The potential size of the associations identified was relatively modest compared to other established modifiable risk factors for lower birth weight and SGA, such as smoking [46,47,48], alcohol [49,50] and caffeine [51,52,53], and small enough to be potentially explained by bias in study design, analysis, or selective publication. There was no evidence of an association with preterm delivery, though only two studies reported results for spontaneous preterm delivery.

Results were broadly consistent across studies collecting urine samples earlier in pregnancy and those collecting later, and across different populations defined by median UIC and geographical region, with adjustment for confounding making little difference too.

Our results are consistent with the most recent review, which found insufficient evidence of an association UIC and low birth weight, but adds to this previous work with findings on birth weight centile and SGA. Furthermore, we identified nonlinearity in some of the dose-response curves and differences between subgroups, none of which would have been apparent without the additional information provided by the more recent cohorts. For example, we saw threshold effects in the associations between I:Cr and birth weight and between iodide intake and birth weight, and noted a potential u-shaped association between I:Cr and SGA.

We have included results from studies reporting iodine exposure as both UIC and I:Cr, with the latter identifying stronger associations with birth outcomes. I:Cr may hold some advantage for epidemiological studies in pregnancy, because UIC cannot adequately estimate iodine status of individuals because of day-to-day variation in iodine status and urine dilution. Results from our review support this, with less between-study heterogeneity with I:Cr, and clearer associations despite the smaller number of studies. However, in addition to maternal kidney function, creatinine excretion may also vary with age, physical activity levels, and body mass index, which themselves may be associated with fetal growth [11].

One well-conducted case-control study nested with the Finnish Medical Birth Register using serum iodide as a measure of iodine status was excluded as outside our pre-defined list of exposures in our protocol [54]. The authors found serum iodide was associated with higher odds of preterm birth, but not with SGA. Serum iodide offers an additional measure of iodine status for consideration. In addition, two studies were excluded for using pre-conception measures of iodine status not specified in our pre-defined protocol [55,56]. However, this timeframe is very relevant and may offer a good measure of maternal iodine resources early in pregnancy.

Taking risk of bias into account, we interpret results for preterm delivery cautiously, because of potential reverse causality, with the decision to deliver influenced by fetal growth up to that point. Otherwise, our assessment of study quality mostly differed by choice of covariate adjustment. Standardised measures of birth weight, such as birth weight centile and SGA controlled for important potential confounding, led to broadly reduced between-study heterogeneity in our results, and potentially provide more informative outcomes for obstetricians and paediatricians.

Our review was partly limited by the available data. As with all observational studies, residual confounding within each study is a potential source of bias, with covariate adjustments differing between studies. Self-reported intake is also prone to measurement error, which can bias associations in either direction. In addition, not all studies reported all birth outcomes or all measures of iodine status. Whilst there was no evidence of small study effects such as publication bias for the more widely reported outcomes of birth weight and preterm delivery, there is more opportunity for publication bias in the less frequently reported outcomes, which we could not assess for small study effects using Egger’s test because of insufficient data.

Very few randomised controlled trials of iodine supplementation or fortification have been conducted outside severely iodine-deficient settings [57], so we have focused on meta-analysis of well-conducted cohort studies as the next strongest evidence. However, without random allocation, our ability to infer causality is limited due to potential residual confounding. All but one study of iodide intake reported only dietary sources, so we included both dietary and total iodide intake in the same meta-analyses. However, because of high supplement use in pregnancy, supplemental sources make a large contribution to total iodine intake [16]. Therefore, studies of dietary intake alone that contribute towards our meta-analyses may have underestimated total intake and overestimated any associations.

Most studies reviewed measured iodine status in spot urines for pragmatic reasons. These are prone to greater variability than 24-h urine collections, so subsequent results are inevitably less precise. Furthermore, the timing of the urine collections was averaged within each study for subgroup comparisons, potentially masking some within-study variation.

Our review is strengthened by the large number of well-conducted birth cohorts that have recently published on this topic allowing greater precision in our results and allowing us to identify potential threshold effects and differences between pre-defined subgroups. We have also been able to explore urine dilution-adjusted results using I:Cr, with novel findings, as well as including self-reported iodide intake. We have conducted a comprehensive review, in terms of the range of different measures of iodine status and birth outcomes that give an overall picture of the association between maternal iodine status and birth outcomes.

## 5. Conclusions

In conclusion, we have found evidence of modest associations between some measures of iodine status, birth weight, birth weight centiles standardised for gestational age. In pregnancy, we recommend reporting iodine status accounting for urine dilution variability using I:Cr as well as the more usual UIC, because of inconsistency in findings between the two measures. Any future studies should also report standardised measures of birth weight for more accurate comparison of outcomes. Whilst the associations are small and not found in all outcomes, they are potentially still important at a population level, and we recommend consideration of methods to maintain iodine sufficiency in the population, especially for women of childbearing age on restricted diets.

## Figures and Tables

**Figure 1 nutrients-15-00387-f001:**
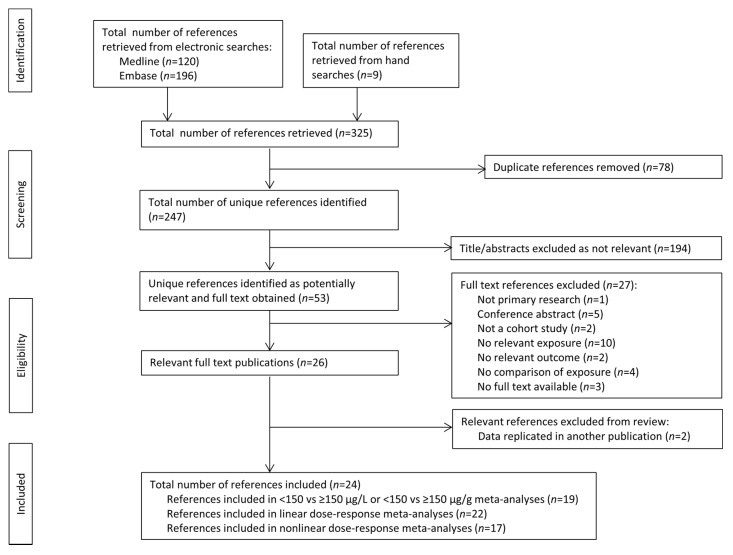
Article retrieval and screening process flow chart.

**Figure 2 nutrients-15-00387-f002:**
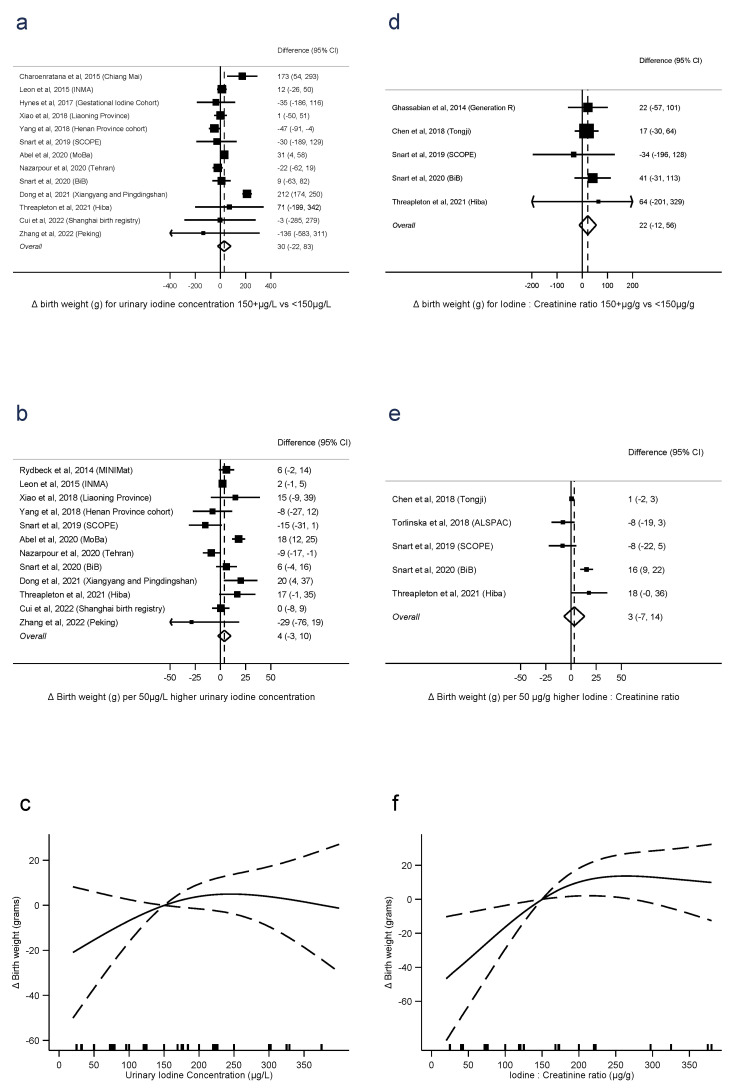
Urinary iodine concentration (**a**–**c**), iodine to creatinine ratio and birth weight (grams) (**d**–**f**), dichotomous (**a**,**d**), linear (**b**,**e**), and nonlinear (**c**,**f**) meta-analyses. CI: Confidence interval; Δ: Change in outcome [11,12,13,14,16,28,29,31,33,34,36,37,39,40,41,42,44].

**Figure 3 nutrients-15-00387-f003:**
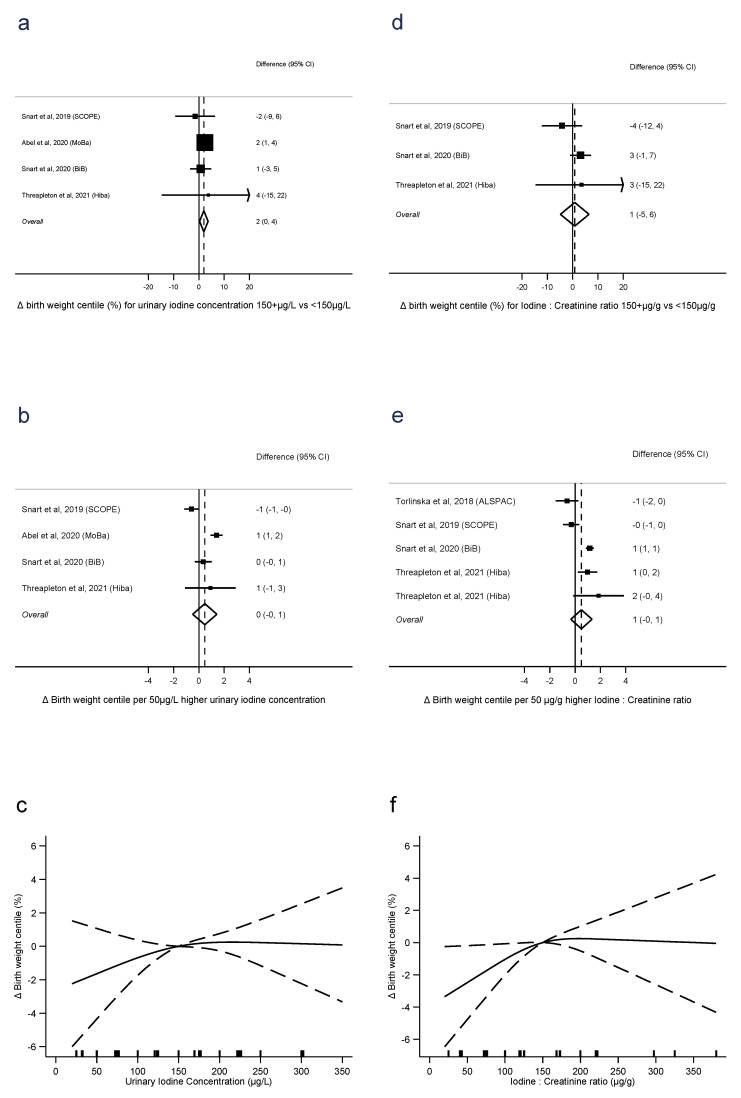
Urinary iodine concentration (**a**–**c**), iodine to creatinine ratio (**d**–**f**) and birth weight centile, dichotomous (**a**,**d**), linear (**b**,**e**), and nonlinear (**c**,**f**) meta-analyses. CI: Confidence interval; Δ: Change in outcome [11,12,13,16,40].

**Figure 4 nutrients-15-00387-f004:**
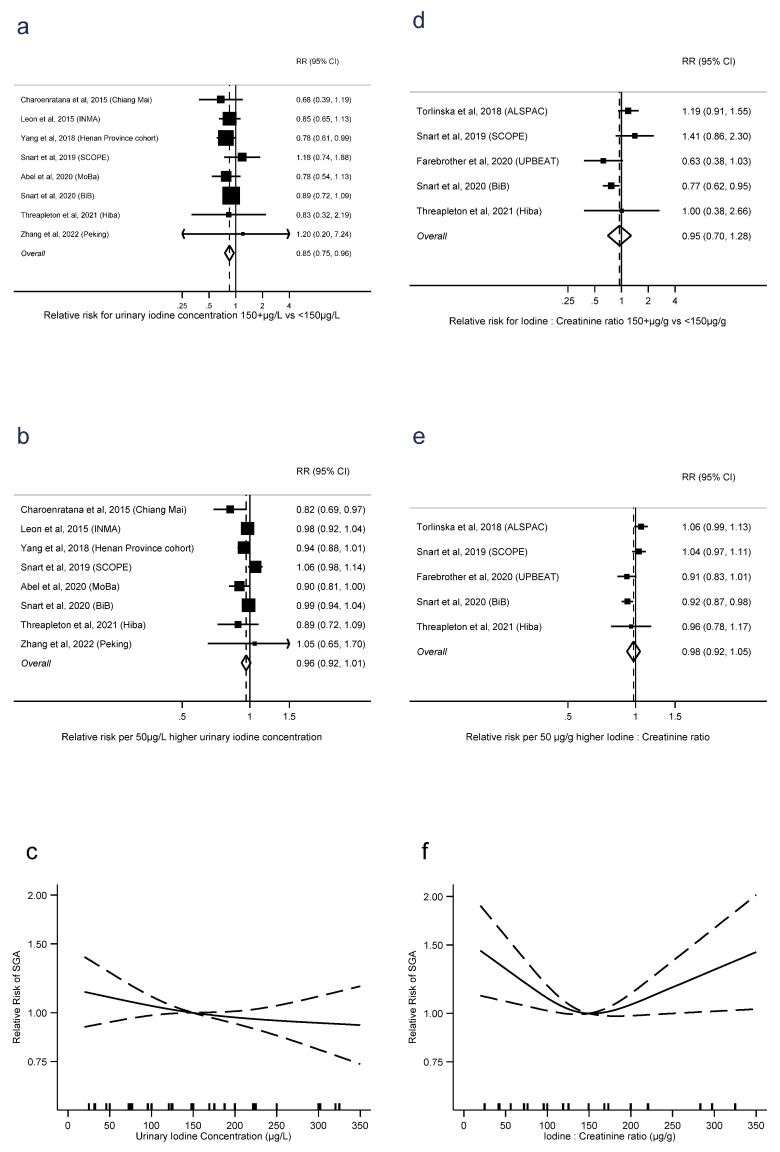
Urinary iodine concentration (**a**–**c**), iodine to creatinine ratio (**d**–**f**) and small for gestational age, dichotomous (**a**,**d**), linear (**b**,**e**) and nonlinear (**c**,**e**) meta-analyses. CI: Confidence interval; RR: Relative risk [11,12,13,16,28,32,36,40,42,44].

**Figure 5 nutrients-15-00387-f005:**
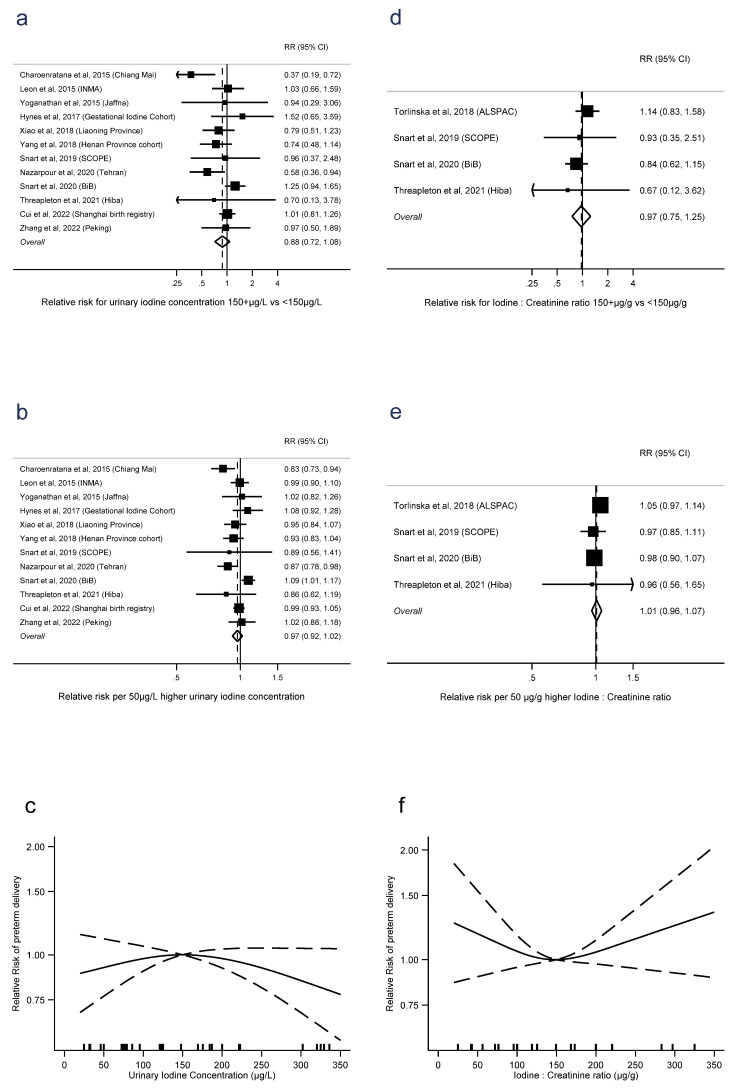
Urinary iodine concentration (**a**–**c**), iodine to creatinine ratio (**d**–**f**) and preterm delivery, dichotomous (**a**,**d**), linear (**b**,**e**), and nonlinear (**c**,**f**) meta-analyses. CI: Confidence interval; RR: Relative risk [12,13,14,16,28,34,36,37,40,41,42,43,44].

**Table 1 nutrients-15-00387-t001:** Characteristics of included studies.

Author, Year	Cohort	Country	Study Size	Trimester	Mean Gestational Age (Weeks)	Median UIC (μg/L)	Exposure Measures	Outcome Measures
Abel, 2020 [11]	MoBa	Norway	2795	2	18.5	69	UIC, dietary intake	birth weight, birth weight centile, SGA, LGA, preterm
Bienertová-Vašků, 2018 [27]	ELSPEC	Czech Republic	4711	3	32	151	dietary intake	birth weight, birth length,
Charoenratana, 2015 [28]	Chiang Mai University cohort	Thailand	399	1–3	19	151	UIC	birth weight, low birth weight, SGA, preterm
Chen, 2018 [29]	Tongji Maternal and Child Health Cohort	China	2087	1–2	13.8	178	I:Cr	birth weight, birth length, head circumference
Cui, 2022 [14]	Shanghai birth registry	China	7435	1–3	18	138	UIC	birth weight, low birth weight, macrosomia, birth length, preterm
Dillon, 2000 [30]	Casamance and Senegal Oriental cohorts	Senegal	462	1–3	19	43	UIC	miscarriage, stillbirth
Dong, 2021 [31]	Xiangyang and Pingdingshan cohorts	China	870	1–3	18	172	UIC	birth weight, macrosomia, birth length, head circumference,
Farebrother, 2020 [32]	UPBEAT	UK	954	2	17	147	I:Cr	birth weight, low birth weight, SGA,
Ghassabian, 2014 [33]	Generation R	The Netherlands	1525	1–2	13.3	119	I:Cr	birth weight
Hynes, 2017 [34]	Gestational Iodine Cohort	Australia	266	2	23.7	83	UIC	birth weight, low birth weight, preterm
Kianpour, 2019 [35]	Isfahan University study	Iran	418	1	9.7	172	UIC	miscarriage
Lean, 2013 [45]	Maharashtra study	India	234	2,3	17, 34	203, 211	UIC	birth weight, birth length
Leon, 2015 [36]	INMA	Spain	2170	1–2	13.4	128	UIC	birth weight, SGA, LGA, preterm
Nazarpour, 2020 [37]	Tehran Thyroid & Pregnancy Study	Iran	1054	1	11	142	UIC	birth weight, low birth weight, birth length, head circumference, preterm
Ovadia, 2022 [38]	Ashkelon study	Israel	134	3	31	61	dietary intake	birth weight, low birth weight, macrosomia, birth weight centile, SGA, LGA, birth length, head circumference, stillbirth, preterm
Rydbeck, 2014 [39]	MINIMat	Bangladesh	1617	1	8	300	UIC	birth weight, birth length, head circumference
Snart, 2019 [12]	SCOPE	UK	541	2	15, 20	134	UIC, I:Cr	birth weight, birth weight centile, SGA, birth length, head circumference, preterm
Snart, 2020 [13]	BiB	UK	6971	2	26	76	UIC, I:Cr	birth weight, birth weight centile, SGA, head circumference, preterm
Threapleton, 2021 [16]	Hiba	UK	246	1, 2, 3	14, 26, 36	122	UIC, I:Cr, dietary intake, total intake	birth weight, low birth weight, birth weight centile, SGA, preterm
Torlinska, 2018 [40]	ALSPAC	UK	1954	1–2	13	95	I:Cr	birth weight, birth weight centile, SGA, LGA, preterm
Xiao, 2018 [41]	Liaoning Province	China	1569	1	7	160	UIC	birth weight, low birth weight, macrosomia, miscarriage, preterm
Yang, 2018 [42]	Henan Province cohort	China	2347	2	27.1	204	UIC	birth weight, low birth weight, macrosomia, SGA, birth length, head circumference, preterm
Yoganathan, 2015 [43]	University of Jaffna study	Sri Lanka	477	3	39.3	140	UIC	preterm
Zhang, 2022 [44]	Peking University International Hospital	China	726	1	6	159	UIC	birth weight, low birth weight, macrosomia, SGA, birth length, preterm

UIC = urinary iodine concentration, I:Cr = urinary iodine to creatinine ratio, SGA = Small for gestational age, LGA = Large for gestational age.

## Data Availability

Example template data collection spreadsheets are available from authors on reasonable request.

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
