# Peer review of "Maternal Iodine Status and Birth Outcomes: A Systematic Literature Review and Meta-Analysis"

_nutrients, 2023, doi:10.3390/nu15020387_

Round 1

Reviewer 1 Report

Dear Authors,

The following minor revisions are recommended for improvement before publication:

Title: There seems to be an error in the title of the manuscript?

Introduction:

lines 48-50: This part is not clear for me. If the consumption of food products rich in iodine increases, the iodine status should improve?

In my opinion this part should be describe some more: it is worth mentioning what are the recommendations regarding iodine intake for pregnant women, what is the scale of iodine deficiency in the world in this population group.

I cannot agree that fish and seafood are the main source of iodine in the diet. In many countries, iodized salt is mandatory, and it is an important source of this nutrient.

Material and methods:

In this part the authors should provide information on the key words used for searching. The detail information is in supplementary files but for the convenience of the reader this information should be in the manuscript.

Results:

From this part, the pages of the manuscript are in landscape order - this makes it difficult to read. There are parts missing in the text: Table 1 – this table is presented after the references?

In my opinion, the graphs should be included in the text of the manuscript. Placing them in additional materials or in the end of the manuscript makes it difficult to understand the content of the manuscript.

It is an interesting and valuable work, but the authors should improve its editing to make it more readable.

Author Response

We thank the reviewers and editorial team for their helpful comments. We address each one in turn below and hope that you agree that this has led to an improved manuscript.

Reviewer #1

  1. Title: There seems to be an error in the title of the manuscript?

Our title is currently “Maternal iodine status and birth outcomes: a systematic literature review and meta-analysis”. We are happy to consider any changes suggested, whilst keeping within the PRISMA and MOOSE guidelines.

  1. Introduction: lines 48-50: This part is not clear for me. If the consumption of food products rich in iodine increases, the iodine status should improve?

Thank you, this paragraph got mangled and the key phrase deleted. The paragraph should say “… recent increases in diets that restrict intake of these foods…”. This has now been corrected.

  1. Introduction: In my opinion this part should be describe some more: it is worth mentioning what are the recommendations regarding iodine intake for pregnant women, what is the scale of iodine deficiency in the world in this population group.

Thank you. We have now included an additional paragraph citing WHO recommendations regarding iodide intake and a recent review of the WHO Global Database on iodine deficiency disorders, which monitors iodide intake across >90% of the world’s population, and a review of iodide intake during pregnancy in the WHO European Region.

  1. Introduction: I cannot agree that fish and seafood are the main source of iodine in the diet. In many countries, iodized salt is mandatory, and it is an important source of this nutrient.

Thank you. We have now changed this to clarify that these are the main dietary sources, excluding supplements and fortified foods. We believe this explains some of the decreases in iodine status seen amongst women of childbearing age in more developed countries without fortification schemes.

  1. Material and methods: In this part the authors should provide information on the key words used for searching. The detail information is in supplementary files but for the convenience of the reader this information should be in the manuscript.

We now provide further detail on key words and phrases used, keeping the full search strategies, synonyms, wildcards and adjacency terms used for each database in supplemental material.

  1. Results: From this part, the pages of the manuscript are in landscape order - this makes it difficult to read. There are parts missing in the text: Table 1 – this table is presented after the references?

We apologise if the pdf provided by the journal did not format our manuscript correctly. I hope the journal editors will be able to provide you with a more readable format if you require it.

  1. Results: In my opinion, the graphs should be included in the text of the manuscript. Placing them in additional materials or in the end of the manuscript makes it difficult to understand the content of the manuscript.

We have provided the main graphs within the main text of the manuscript in figures 1 to 5, which is the maximum number of figures we can include. Figure 1 contains the important PRISMA flowchart. The remaining 4 figures each contain a panel of 6 sub-figures, presenting all the main results for the four outcomes and two exposures with the most evidence found in the literature. The supplemental material contains the graphs for the remaining outcomes and subgroup analyses.

  1. It is an interesting and valuable work, but the authors should improve its editing to make it more readable.

Thank you for the helpful suggestions. We hope that these additions and clarifications improve the manuscript.

Reviewer 2 Report

It is an interesting meta analysis on iodine status in pregnancy and both fetal and maternal outcomes. A few points should be addressed. 

Introduction:

  • Lines 48-50: “Dairy and seafood are the main sources of dietary iodine. With recent increases in 48 diets that these foods, particularly in women of childbearing age, the iodine status of populations previously considered sufficient are moving more towards mild deficiency” this sentence is not clear and needs to be re-written.

  • Lines 53-55: “Since the most recent systematic reviews of observational studies, which drew different conclusions from the same the same evidence, results on iodine and birth outcomes from a number of large birth cohorts have been published.”: the sentence is not clear, need revision

Methods:

  • Line 89: Titles and abstracts were screened by two independent reviewers (CK, ET, JW, LJH). Why are there 4 acronyms?

Discussion:

  • I think that the result obtained on the association between SGA and both low and HIGH I:Cr deserves a comment, to warn about not only the risks related to poor iodine intake, but also the risks related to an excess of Iodine/more than adequate status. 

Author Response

We thank the reviewers and editorial team for their helpful comments. We address each one in turn below and hope that you agree that this has led to an improved manuscript.

Reviewer #2

  1. It is an interesting meta analysis on iodine status in pregnancy and both fetal and maternal outcomes. A few points should be addressed. 

Thank you.

  1. Introduction: Lines 48-50: “Dairy and seafood are the main sources of dietary iodine. With recent increases in 48 diets that these foods, particularly in women of childbearing age, the iodine status of populations previously considered sufficient are moving more towards mild deficiency” this sentence is not clear and needs to be re-written.

Thank you, this paragraph got mangled and the key phrase deleted. The paragraph should say “… recent increases in diets that restrict intake of these foods…”. This has now been corrected.

  1. Introduction: Lines 53-55: “Since the most recent systematic reviews of observational studies, which drew different conclusions from the same the same evidence, results on iodine and birth outcomes from a number of large birth cohorts have been published.”: the sentence is not clear, need revision

Thank you. We have split up the long sentence into two shorter sentences. We have also added further text and references to clarify the meaning.

  1. Methods: Line 89: Titles and abstracts were screened by two independent reviewers (CK, ET, JW, LJH). Why are there 4 acronyms?

We divided the work up between the team. Each article was independently screened by two reviewers from the list of four. We have now clarified this in the text.

  1. Discussion: I think that the result obtained on the association between SGA and both low and HIGH I:Cr deserves a comment, to warn about not only the risks related to poor iodine intake, but also the risks related to an excess of Iodine/more than adequate status. 

Thank you for this observation. We have added a brief comment on this to the 4th paragraph in the discussion, alongside other commentary on threshold effects presenting as nonlinearity in the figures.